# Cell Type Specific Suppression of Hyper-Recombination by Human RAD18 Is Linked to Proliferating Cell Nuclear Antigen K164 Ubiquitination

**DOI:** 10.3390/biom15010150

**Published:** 2025-01-20

**Authors:** Colette B. Rogers, Wendy Leung, Ryan M. Baxley, Rachel E. Kram, Liangjun Wang, Joseph P. Buytendorp, Khoi Le, David A. Largaespada, Eric A. Hendrickson, Anja-Katrin Bielinsky

**Affiliations:** 1Department of Biochemistry, Molecular Biology, and Biophysics, University of Minnesota, Minneapolis, MN 55455, USA; 2Departments of Pediatrics and Genetics, Cell Biology, and Development, University of Minnesota, Minneapolis, MN 55455, USA; 3Department of Biochemistry and Molecular Genetics, University of Virginia, Charlottesville, VA 22903, USA

**Keywords:** gap-filling, hyper-recombination, PCNA K164, RAD18, ubiquitination

## Abstract

RAD18 is a conserved eukaryotic E3 ubiquitin ligase that promotes genome stability through multiple pathways. One of these is gap-filling DNA synthesis at active replication forks and in post-replicative DNA. RAD18 also regulates homologous recombination (HR) repair of DNA breaks; however, the current literature describing the contribution of RAD18 to HR in mammalian systems has not reached a consensus. To investigate this, we examined three independent *RAD18*-null human cell lines. Our analyses found that loss of RAD18 in HCT116, but neither hTERT RPE-1 nor DLD1 cell lines, resulted in elevated sister chromatid exchange, gene conversion, and gene targeting, i.e., HCT116 mutants were hyper-recombinogenic (hyper-rec). Interestingly, these phenotypes were linked to RAD18’s role in PCNA K164 ubiquitination, as HCT116 *PCNA^K164R/+^* mutants were also hyper-rec, consistent with previous studies in *rad18*^−/−^ and *pcna^K164R^* avian DT40 cells. Importantly, the knockdown of UBC9 to prevent PCNA K164 SUMOylation did not affect hyper-recombination, strengthening the link between increased recombination and RAD18-catalyzed PCNA K164 ubiquitination, but not K164 SUMOylation. We propose that the hierarchy of post-replicative repair and HR, intrinsic to each cell type, dictates whether RAD18 is required for suppression of hyper-recombination and that this function is linked to PCNA K164 ubiquitination.

## 1. Introduction

DNA damage and replication stress can cause the accumulation of single-stranded (ss) DNA gaps in nascent DNA. To avoid the deleterious effects of these gaps on genome stability and cell fitness, cells rely on DNA damage tolerance and repair pathways. These include gap-filling synthesis that occurs “on the fly” at replication forks [1] as the replisome proceeds—and has been reconstituted in vitro [2]—as well as post-replicative synthesis pathways [2,3]. If these mechanisms fail, the resulting DNA gaps or breaks may be repaired with high fidelity through homologous recombination (HR) [4,5]. Two protein factors integral to many of these pathways are the E3 ubiquitin ligase radiation sensitive 18 (RAD18) and the replicative sliding clamp, proliferating cell nuclear antigen (PCNA).

The functional interplay between RAD18 and PCNA has been best characterized in the context of translesion DNA synthesis (TLS), an evolutionarily conserved gap-filling synthesis pathway that may function “on the fly” at replication forks or in post-replicative regions. To activate TLS, RAD18 mono-ubiquitinates PCNA at the conserved lysine 164 (K164) residue, which serves to recruit TLS polymerases to replace stalled replicative polymerases [6,7], to facilitate the bypass of DNA lesions [8] and prevent ssDNA accumulation. Once lesion bypass is complete, the replisome resumes DNA synthesis utilizing replicative polymerases to complete DNA replication. In the absence of productive TLS, mono-ubiquitinated PCNA K164 can be subsequently modified by K63-linked polyubiquitin chains [9,10,11,12] to promote template switching (TS), an HR-based mechanism where the undamaged nascent strand of the sister chromatid is temporarily utilized as a template to circumvent the lesion [6,13,14]. Notably, PCNA K164 is also modified by the addition of a small ubiquitin-like modifier (SUMO) [6,7,15] mediated by the E2 conjugating enzyme ubiquitin carrier protein 9 (UBC9) [16]. This SUMOylation suppresses HR at replication forks and in post-replicative DNA [17,18], while also indirectly promoting damage bypass and gap-filling synthesis by enhancing the ubiquitin ligase activity of RAD18 [16,19].

Repair processes that operate outside of the S phase also utilize RAD18 and PCNA K164 activities to promote the completion of DNA synthesis. Break-induced replication (BIR) is a conservative DNA synthesis pathway that utilizes strand invasion in regions of short homology to complete repair synthesis of under-replicated DNA [20]. A similar mechanism is used during mitotic DNA synthesis (MiDAS) as cells make a final attempt at duplicating unreplicated parental DNA located between stalled replication forks [21,22], and at telomeres maintained by the alternative lengthening of telomeres (ALT) pathway [23,24,25]. Interestingly, the molecular mechanisms and machinery involved in these post-replicative synthesis pathways, including RAD18 catalyzed PCNA K164 ubiquitination, straddle the divide between “on the fly” gap-filling processes at the replication fork and classical homology-dependent double-strand break (DSB) repair.

Failure to complete gap-filling repair synthesis can result in fork collapse and DNA breaks that are ideally repaired with high fidelity through HR. Research to date has clearly demonstrated a role for RAD18 in HR, although its contribution appears species and/or context dependent. In budding yeast, loss of *RAD18* led to hyper-recombinogenic (hyper-rec) phenotypes, including higher rates of spontaneous interchromosomal recombination [26], elevated levels of spontaneous recombination, ectopic gene conversion, recombination between direct repeats [27], and enhanced gross chromosomal rearrangements [28]. Consistent with these observations, avian DT40 *rad18* mutant cells exhibited elevated levels of sister chromatid exchange (SCE), which involves recombination between two sister DNA duplexes [29,30,31]. There are conflicting reports on the effects of RAD18 on recombination in mammalian cells. Human HCT116 *RAD18*^−/−^ cells had a nearly 10-fold increase in HR-dependent gene targeting [32], and *RAD18* mutant mouse embryonic stem cells exhibited elevated levels of SCE and targeted integration [33], suggesting that mammalian RAD18 suppresses recombination as it does in yeast and chicken cells. In contrast, it has been reported that RAD18 promotes HR by recruiting radiation sensitive 51C (RAD51C) to DSBs [34]. Consistent with this, overexpression of *RAD18* increased CRISPR/Cas9-mediated HR activity in human cells [35,36].

The discrepant outcomes of depleting RAD18 on HR activity in mammalian cells prompted us to examine HR-mediated repair in three model human cell lines. HCT116 *RAD18*^−/−^ cells exhibited hyper-rec phenotypes including elevated SCE, plasmid-templated gene conversion, and CRISPR/Cas9-mediated targeted integration. However, these hyper-rec phenotypes were not observed in DLD1 or hTERT RPE-1 *RAD18*^−/−^ cell lines and could not be explained by differences in the mismatch repair (MMR) status of the cell lines. Consistent with their published roles, knock-out of *RAD18* or a mono-allelic knock-in of a *PCNA^K164R^* mutation in HCT116 cells led to defective UV-induced PCNA mono-ubiquitination and TLS. Interestingly, HCT116 *PCNA^K164R/+^* mutants, which can only ubiquitinate one half of their cellular PCNA protein showed elevated SCE both genome-wide and specifically at telomeres, which are known to be hard to replicate. Moreover, hyper-recombination in *PCNA^K164R/+^* mutants was not rescued by knockdown of the SUMO conjugating enzyme UBC9, showing that hyper-recombination is linked to K164 ubiquitination but not SUMOylation. Taken together, our data suggest that HR may be upregulated in HCT116 cells (and by extension in certain other human cell types) to compensate for reduced activity of gap-filling DNA synthesis pathways, e.g., TLS, which are linked to RAD18-dependent PCNA ubiquitination.

## 2. Materials and Methods

### 2.1. Cell Culture

HCT116 cells (CCL-247; RRID:CVCL_0291) were grown in McCoy’s 5A medium (Corning, Corning, NY, USA; 10-050-CV) supplemented with 10% fetal bovine serum (FBS, Sigma, Burlington, MA, USA; F4132), 1% penicillin-streptomycin (Pen-Strep, Gibco, Life Technologies, Carlsbad, CA, USA; 15140), and 1% L-glutamine (L-Gln, Gibco, Life Technologies, Carlsbad, CA, USA; 205030). DLD1 cells (CCL-221; RRID: CVCL_0248) were grown in RPMI 1640 medium (Gibco, Life Technologies, Carlsbad, CA, USA; A10491), 10% FBS, 1% Pen-Strep, and 1% L-Gln. hTERT RPE-1 (CRL-4000; RRID:CVCL_4388) cells were grown in DMEM/F12 medium (Gibco, Life Technologies, Carlsbad, CA, USA; 11320), 10% FBS, and 1% Pen-Strep. All cells were cultured at 5% CO_2_ and 37 °C. Cell lines used in this study were authenticated using several methods including morphology checks by microscope, growth curve analyses, PCR and Sanger sequencing, mycoplasma testing, and high-quality G-band karyotype analyses.

### 2.2. Cell Line Generation

To generate HCT116 [37], hTERT RPE-1 [22], and DLD1 [37] *RAD18*^−/−^ cell lines, a synthetic guide RNA (sgRNA) targeting *RAD18* exon 2 (AGACAATAGATGATTTGCTG) was cloned into hSpCas9(BB)-2A-GFP (PX458, Addgene 48138). Parental cell lines were transfected using electroporation (Neon Transfection System MPK5000) and GFP-expressing cells were collected by flow cytometry and single-cell subcloned to isolate clonal populations. Inactivation of *RAD18* was confirmed via Sanger sequencing (RAD18ex2_PCR_Forward: GTAGTACCATGCCGAAAGCAC; RAD18ex2_PCR_Reverse: GGAACCACCTATCTGTTATCC; RAD18ex2_Seq_Forward: TGCAGTTTATCTGGAGTTAGC) and Western blot analyses. Sizes of the INDELs at the Cas9 cute site carried by each clonal *RAD18*^−/−^ cell line are as follows: HCT116 clone 1A3 (−1, −17), clone 1B10 (−10,−10); hTERT RPE−1 clone 1A11 (+2, −17), clone 1B4 (+1,+1); DLD1 clone C2 (−1,−1), clone C4 (−2, −2), clone C5 (−1,−1).

To generate hTERT RPE-1 *MLH1*^−/−^ cell lines, a sgRNA (Synthego Corporation, Redwood City, CA, USA; MLH1ex12: ATTTAACCATCTCCCCAGAG, 100 pmol) and *Cas9* mRNA (TriLink BioTechnologies, San Diego, CA, USA; L-7206) were transfected into parental cell lines and inactivation was confirmed in clonal populations via Sanger sequencing (MLH1ex12_PCR_Forward: AATGGGCACAGAGTTTCAGTTTGG; MLH1ex12_PCR_Reverse: TCAAAGGACACTATGGTGGGTGAC; MLH1ex12_Seq_Reverse: CTGTCTTATCCTCTGTGACAATGG) and Western blot analyses.

To generate HCT116 and hTERT RPE-1 *PIGA^−/o^* cell lines, a sgRNA (Synthego Corporation, Redwood City, CA, USA; PIGAex6: GGTATATGACCGGGTATCAG) and *Cas9* mRNA were transfected into parental cell lines and inactivation was confirmed in clonal populations via Sanger sequencing (PIGAex6_PCR_Forward: CTGTTCAGACTTTGCGGGACTTTG; PIGAex6_PCR_Reverse: CCTTCAGCACCCAAAGCTCTC; PIGAex6_Seq_Forward: GGTGTCTTGCCCAGAATAAGTAC).

To generate *PCNA^K164R/+^* HCT116 cell lines, sgRNAs targeting *PCNA* exon 5 (PCNAex5_gRNA_1: ATACGTGCAAATTCACCAGA; PCNAex5_gRNA_2: GCAAGTGGAGAACTTGGAAA) were cloned into hSpCas9(BB)-2A-GFP (PX458, Addgene 48138). Double-stranded donor plasmids containing the K164R coding mutation (c.491 A>G) were constructed as described [38,39]. Silent coding mutations were introduced to generate novel restriction enzyme recognition sites. HCT116 wild-type cells were electroporated with CRISPR/Cas9 plasmid and donor plasmid (Neon Electroporation System MPK5000). Next, GFP-expressing cells were collected by flow cytometry and inactivation was confirmed in clonal populations via Sanger sequencing and restriction enzyme digestion (PCNAex5_F: TGGCGCTAGTATTTGAAGCA, PCNAex5_R: ACTTGGGATCCAATTCTGTCTACT, restriction enzyme for PCNAex5_gRNA_1: EcoRI, NEB, Ipswich, MA, USA; R3101; restriction enzyme for PCNAex5_gRNA_2: XcmI, NEB, Ipswich, MA, USA; R0533). Mutations were validated by Sanger sequencing (PCNAex5_Seq: AGGTGTTGCCTTTTAAGAAAGTGAGG).

To generate reverted *PCNA^K164/K164^* cell lines from *PCNA^K164R/+^* mutant cells, a sgRNA (5′-GCTAGTGGAGAACTTGGAAA-3′) targeting the R164-allele of *PCNA* exon 5 was cloned into hSpCas9(BB)-2A-GFP (PX458, Addgene 48138). A double-stranded donor was constructed with a point mutation designed to revert the arginine (AGA) to lysine (AAA) coding mutation (c.491 A>G) back to c.491A and silent mutations were introduced to remove the EcoRI restriction enzyme recognition site while three silent downstream mutations were introduced to generate a novel XcmI restriction enzyme recognition site. Cells were electroporated with CRISPR/Cas9 and donor plasmid (Neon Electroporation System MPK5000). Knock-in cell lines containing the desired modifications were identified in clonal populations by PCR (PCNAex5_F: TGGCGCTAGTATTTGAAGCA, PCNAex5_R: ACTTGGGATCCAATTCTGTCTACT), followed by restriction enzyme digestion and Sanger sequencing (PCNAex5_Seq: AGGTGTTGCCTTTTAAGAAAGTGAGG).

To generate HCT116 *PCNA^K164R/+^* cell lines expressing Super-Telomerase (ST), pBABEpuroUThTERT+U3-hTR-500 (Addgene 27665) [40,41] was linearized with restriction enzyme ScaI (NEB, Ipswich, MA, USA; R3122) and transfected into cell lines following standard Lipofectamine 3000 protocols (Invitrogen, Thermo Fisher Scientific, Waltham, MA, USA; L3000). Stable cell lines were generated using puromycin selection (1 µg/mL; Sigma, Burlington, MA, USA; P7255) followed by subcloning.

HCT116 *RAD18*^−/−^ clone 1A3 was complemented with a FLAG-tagged *RAD18* cDNA cloned into a PiggyBac transposition vector containing a cytomegaloviral/*beta*-actin/*beta*-globin (CAG) promoter and a neomycin resistance selection cassette. One million *RAD18*^−/−^ cells were transfected with 1 μg of the *RAD18* cDNA-containing vector and 1 μg of a PiggyBac transposase expression vector (Neon Electroporation System MPK5000). Individual neomycin-resistant clones were isolated and analyzed for RAD18-FLAG expression via Western blot analyses.

### 2.3. Western Blot Analyses

For preparation of whole cell extracts, cells were lysed with NETN buffer (20 mM Tris-HCl, pH 8, 100 mM NaCl, 1 mM EDTA, 0.5% NP-40, and protease inhibitors) for 10 min at 4 °C and centrifuged at 12,000 rpm at 4 °C. Bradford assays (Bio-Rad Laboratories, Hercules, CA, USA) were performed to determine protein concentration. For preparation of chromatin fractions, cells were lysed with Buffer A (10 mM HEPES pH 7.9, 10 mM KCl, 1.5 mM MgCl_2_, 0.34 M sucrose, 10% glycerol, 0.1% Triton X-100, and protease inhibitors) for 5 min at 4 °C and then centrifuged at 1300× *g* at 4 °C. The soluble fractions were removed, and nuclear pellets were resuspended in TSE buffer (20 mM Tris-HCl, pH 8, 500 mM NaCl, 2 mM EDTA, 0.1% SDS, 0.1% Triton X-100, and protease inhibitors), sonicated, and centrifuged at 17,000× *g* at 4 °C. Bradford assays were performed to determine protein concentration. Extracts were mixed with SDS loading buffer, denatured at 99 °C, electrophoresed on SDS-PAGE gels, and transferred to nitrocellulose membranes. Membranes were blocked in 5% milk (G-Biosciences, St. Louis, MO, USA; 786-011) for 1 h and incubated in primary antibodies overnight at 4 °C. Primary antibodies were diluted in 5% milk as follows: rabbit anti-RAD18 (Bethyl Laboratories, Montgomery, TX, USA; A300-340A, RRID:AB_937974; 1:1000), mouse anti-PCNA (Abcam, Cambridge, UK; ab29, RRID:AB_303394; 1:3000), rabbit anti-ubiquityl-PCNA (Lys164) (Cell Signaling Technology, Danvers, MA, USA; D5C7P, 13439, RRID:AB_2798219; 1:1000), rabbit anti-MLH1 (Abcam, Cambridge, UK; ab92312, RRID:AB_2049968; 1:1000), mouse anti-FLAG (Sigma, Burlington, MA, USA; F3165, RRID:AB_259529; 1:1000), mouse anti-α-tubulin (Millipore-Sigma, Burlington, MA, USA;, T9026, clone DM1A, RRID:AB_477593; 1:10000), anti-Pol η (Abcam, Cambridge, UK; ab234855, RRID:AB_3094753; 1:1000), rabbit anti-H2AX (Bethyl Laboratories, Montgomery, TX, USA; A300-082A, RRID:AB_203287; 1:5000), rabbit anti-γH2AX (Bethyl Laboratories, Montgomery, TX, USA; A300-081A, RRID:AB_203288; 1:2000), rabbit anti-RPA32 (phospho S4/8) (Bethyl Laboratories, Montgomery, TX, USA; A300-245A, RRID:AB_210547; 1:2000), anti-β-actin (Novus Biologicals, Centennial, CO, USA; NB600-501, RRID_AB:10077656, 1:10000), and anti-UBC9 (Santa Cruz Biotechnology, Dallas, TX, USA; C-12, sc-271057, RRID:AB_10610674; 1:500). Membranes were incubated in secondary antibodies diluted in 5% milk: goat anti-mouse HRP conjugate (Jackson Laboratories, Bar Harbor, ME, USA; 115-035-003, RRID:AB_10015289; 1:10000) or goat anti-rabbit HRP conjugate (Jackson Laboratories, Bar Harbor, ME, USA; 111-035-144, RRID:AB_2307391; 1:10000). Alternatively, blocking, primary and secondary antibody incubations using rabbit anti-ubiquityl-PCNA (Lys164) (Cell Signaling Technology, Danvers, MA, USA; D5C7P, 13439, RRID:AB_2798219; 1:1000) were performed using 5% BSA in TBST (0.1% Tween 20). Detection was performed using WesternBright Quantum detection (K-12042-D20) and quantification was performed using the FIJI version 2.14.0/1.54f and Microsoft Excel version 16.93.

### 2.4. Clonogenic Survival Assay

HCT116 cells were seeded at 500 cells or 800 cells (*PCNA^K164R/+^*) per well in triplicate in 6-well plates. After 24 h, the medium was removed and wells were either washed gently with prewarmed 1× PBS and exposed to UV (UVP CL1000), or the medium was removed and replaced with MMS-containing medium (Acros Organics, Thermo Fisher Scientific, Waltham, MA, USA; 156890250). Cells were incubated for 10 to 14 days, washed in 1× PBS, fixed in 10% acetic acid/10% methanol, and stained with crystal violet. Colonies reaching a minimum size of 50 cells were counted manually and normalized to the average colony number in untreated wells.

### 2.5. Direct-Repeat (DR)-Green Fluorescent Protein (GFP) Gene Conversion Assay

The DR-GFP gene conversion assay was performed as described [42]. One million cells were electroporated (Neon Electroporation System MPK5000) with 0.66 μg of each of the following plasmids: (i) pDRGFP (Addgene 26475), (ii) pCBASceI (Addgene 26477), and (iii) pCAGGS-mCherry (Addgene 41583). Transfected cells were seeded in triplicate in 6-well plates and allowed to recover in culture for 72 h. After three days, cells were harvested and pelleted by centrifugation, washed once with 1× PBS, and fixed for 30 min in 500 uL 3.7% formaldehyde/PBS dropwise while vortexing gently. After fixation, cells were pelleted by centrifugation, washed with 1× PBS twice, and analyzed via FACS using a BD Fortessa X-20. The percentage of gene conversion was calculated by measuring the percentage of mCherry-positive cells that were GFP-positive.

### 2.6. Phosphatidylinositol Glycan Class A (PIGA) Gene Targeting Assay

One million HCT116 or RPE-1 *PIGA^−/0^* cells were electroporated (Neon Electroporation System MPK5000) with (i) 100 pmol of a sgRNA targeting *PIGA* intron 5 (TGGGTGAAAGTGCTCACACT), (ii) 1 μg *Cas9* mRNA, (iii) 1 μg *PIGA* donor plasmid harboring the corrected *PIGA* exon 6 sequence, and (iv) 1 μg mCherry expression control plasmid. Transfected cells were seeded in triplicate in 6-well plates and allowed to recover in culture for 72 h. After three days, cells were harvested, pelleted by centrifugation, washed once with 1× PBS, and fixed for 30 min in 500 uL 3.7% formaldehyde/PBS dropwise while vortexing gently. After fixation, cells were pelleted by centrifugation, washed once with 1x PBS, and stained with 5 nM Alexa Fluor 488 proaerolysin (FLAER) (Cedarlane, FL1S) in PBS for 30 min at room temperature in the dark. Finally, stained cells were pelleted by centrifugation, washed once with 1× PBS, and FACS analysis was performed using the BD Fortessa X-20 cytometer. The percentage of *PIGA* gene targeting was calculated by measuring the percentage of mCherry-positive cells that were Alexa Fluor 488-positive.

### 2.7. Hypoxanthine Phosphoribrosyltransferase (HPRT) Mutagenesis Assay

HCT116 *HPRT^+/o^* cells were selected by maintaining cells in culture for at least two weeks in a medium supplemented with hypoxanthine-aminopterin-thymidine (HAT) (Sigma, Burlington, MA, USA; H0262). After HAT selection, cells were seeded at 15,000 cells per well in 6-well plates. After 24 h, cells were untreated or treated with 6 J UV and cultured for one week. Next, cells were seeded for colony formation assays (50,000 cells in triplicate in 6-well plates). Additionally, cells were seeded to measure plating efficiency (500 cells in duplicate in 6-well plates). Then, 24 h after seeding for colony formation assays, cells were treated with 6-thioguanine (6-TG; Sigma, Burlington, MA, USA; A4882; 10 μg/mL). Plating efficiency control plates were not treated with 6-TG. Colonies were grown for approximately 9 days, the medium was removed, and colonies were fixed in 10% acetic acid/10% methanol and stained with crystal violet. Plating efficiency = (average number of colonies)/500 cells. Mutation frequency = (average number of colonies)/(plating efficiency × 50,000 cells).

### 2.8. Small-Interfering RNA (siRNA) Transfection

HCT116 wild-type, *PCNA^K164R/+^,* and *PCNA^K164^* reverted ST cells were seeded in 6-well plates and allowed to recover for 24 h. Cells were transfected with 10 mM *UBC9* siRNA (Horizon Discovery, Waterbeach, UK; ON-TARGETPlus Human UBE2I (7329) siRNA SMARTPool, L-004910-00-0010) using DharmaFECT (Horizon Discovery, Waterbeach, UK; T-2001) for 48 h. A second round of transfection after 24 h was performed to achieve complete knockdown.

### 2.9. Sister Chromatid Exchange (SCE) Assay

Cells were arrested at the G1/S phase boundary using a double thymidine block (Sigma, Burlington, MA, USA; T9250, 2 mM) and released into a fresh complete medium containing 10 μM EdU for one round of replication (~9 h). Next, cells were washed with prewarmed (37 °C) 1× PBS twice and cultured in a complete medium without EdU for 18 h. Three hours prior to collection, cells were treated with 10 µg/mL colcemid (Gibco, Life Technologies, Carlsbad, CA, USA; 15212-012). Cells were harvested, and centrifuged, and pellets were resuspended in 75 mM KCl at room temperature for 30 min. Next, 1 mL of fixative (3:1; methanol:glacial acetic acid) was added dropwise while vortexing gently. Cells were incubated for 10 min and pelleted by centrifugation at 500× *g*. Cell pellets were resuspended with 5 mL fixative dropwise while vortexing gently, incubated at room temperature for 10 min, and centrifuged at 500× *g* three times. Once fixed, ~15 µL of cells were dropped onto prewarmed (37 °C) glass slides and allowed to dry overnight. Next, the slides were rehydrated in 1× PBS, fixed in 3.7% formaldehyde for 10 min, and subjected to a click-chemistry reaction at room temperature in the dark for at least 1 h (1 mM biotin-azide, 100 mM ascorbic acid, and 1 mM CuSO_4_ in PBS). Next, the slides were blocked in ABDIL (20 mM Tris pH 7.5, 2% BSA, 0.2% fish gelatin, 150 nM NaCl, 0.1% NaN_3_) for 1 h and incubated with an anti-streptavidin Alexa Fluor 488 conjugate (Thermo Fisher Scientific, Waltham, MA, USA;, S32354, 1:1000 in ABDIL) for 1 h in the dark at room temperature. Finally, slides were washed three times with PBST (PBS + 0.1% Tween 20) with DAPI added to the second wash (Life Technologies, Carlsbad, CA, USA; D13006) and mounted with Vectashield (Vector Laboratories, Newark, CA, USA; H-1000). Images were taken using a Zeiss (Oberkochen, Germany) Spinning Disk confocal microscope at the University of Minnesota Imaging Center and SCE events were scored double-blinded, manually.

### 2.10. Telomeric SCE (t-SCE) Assay

For analyses of t-SCE, HCT116 ST cells were cultured in the presence of BrdU:BrdC with a final concentration of 7.5 mM BrdU (MP Biomedicals, Irvine, CA, USA; 100166) and 2.5 mM BrdC (Sigma, Burlington, MA, USA; B5002) for 12 h prior to harvesting. Colcemid (Gibco, Life Technologies, Carlsbad, CA, USA, 15212-012) was added at a concentration of 0.1 μg/mL during the last two hours. CO-FISH was performed as described [43] using a TelC-Alexa488-conjugate probe (PNA Bio, Thousand Oaks, CA, USA; F1004), followed by a TelG-Cy3-conjugated probe (PNA Bio, Thousand Oaks, CA, USA; F1006). Images were captured using a Zeiss (Oberkochen, Germany) Spinning Disk confocal microscope and t-SCE events were scored double-blinded, manually.

### 2.11. Telomere Restriction Fragment (TRF) Analysis

Genomic DNA was extracted from ~1 × 10^7^ cells using a modified version of the Gentra Puregene Cell Kit cell extraction protocol (Qiagen, Hilden, Germany; 158745). The integrity of genomic DNA and absence of contaminating RNA were confirmed via 1% agarose 1X TAE gel electrophoresis. Subsequently, 30 to 40 μg of genomic DNA was digested with HinfI (NEB, Ipswich, MA, USA; R0155) and RsaI (NEB, Ipswich, MA, USA; R0167) restriction enzymes, as described [44]. For each sample, 8 to 12 μg of digested genomic DNA was resolved overnight on a 0.7% agarose 1X TBE gel. Gels were depurinated, denatured, and neutralized, followed by overnight capillary transfer to a Hybond-XL membrane (GE Healthcare RPN303S). The telomere probe was radiolabeled using T4 polynucleotide kinase (NEB, Ipswich, MA, USA; M0201) and γ-P^32^-ATP (Perkin Elmer, Waltham, MA, USA; NEG035C) and purified using quick spin columns (Roche, Basel, Switzerland; 11-273-949-001). Membranes were pre-hybridized for 1 h with Church buffer at 55 °C, then hybridized with a γ-P^32^-end-labeled telomere probe (C_3_TA_2_)_4_ in Church buffer at 55 °C overnight. Membranes were washed three times with 4× SSC and once with 4× SSC + 0.1% SDS, each for 30 min, exposed to a phosphorimaging screen, detected with a Typhoon FLA 9500 imager, and processed using FIJI version 2.14.0/1.54f and Adobe Photoshop 25.11.0.

### 2.12. Quantification and Statistical Analysis

GraphPad Prism version 9.0.0 software was used to perform statistical analysis and specifics of the statistical analyses used for each experiment are described in the figure legends. Statistical significance was denoted as follows: ns = not significant; * < 0.05; ** < 0.01; *** < 0.001; **** < 0.0001.

## 3. Results

### 3.1. Hyper-Rec Phenotypes Following the Loss of Human RAD18 Are Cell-Type Specific

To understand the role of human RAD18 in regulating HR-mediated processes, we utilized clonal hTERT RPE-1 [22], DLD1 [37], and HCT116 [37] *RAD18*^−/−^ cell lines. We confirmed the loss of RAD18 expression in these cell lines using Western blot analyses (Figure 1A–C). In wild-type cells, RAD18 was detected as two major bands, consistent with the presence of unmodified and ubiquitinated RAD18 in human cells [45], which were absent in *RAD18*^−/−^ cells. Additionally, the anti-RAD18 antibody detected a non-specific band in wild-type and *RAD18*^−/−^ cell lines, similar to previous observations [46,47]. Thus, as expected, *RAD18*^−/−^ cells do not express RAD18 protein. Next, we performed SCE assays, which measure crossover recombination between sister chromatids in a replicating chromosome (Figure 1D,E) in unperturbed RAD18-proficient and RAD18-deficient cells. The loss of RAD18 did not alter the level of SCEs in either RPE-1 or DLD1 cell lines (Figure 1F,G). However, in HCT116 cells, SCEs approximately doubled (Figure 1H) suggesting that *RAD18*^−/−^ HCT116 cells are hyper-rec. In agreement with this conclusion, the expression of FLAG-tagged RAD18 (at levels lower than endogenous RAD18) rescued the increased SCE phenotype (Figure 1I,J).

Next, we explored the potential link between the RAD18 hyper-rec phenotypes and mismatch repair (MMR) status, as MMR suppresses recombination between homeologous chromosomes [48]. HCT116 cells carry a hemizygous nonsense mutation in the gene encoding the MMR factor MutL homolog 1 (MLH1) [49], which renders the cells MMR-deficient. To test whether loss of MLH1, in general, causes human *RAD18*^−/−^ cells to become hyper-rec, we functionally inactivated *MLH1* in our RPE-1 cell lines (Appendix A). However, neither *MLH1*^−/−^ nor *RAD18*^−/−^*/MLH1*^−/−^ RPE-1 cells exhibited an increase in SCEs (Appendix A). This result is consistent with the fact that DLD1 cells are, like HCT116 cells, also MMR-deficient [50] {due to a MutS homolog 6 (MSH6) deficiency rather than MLH1} and are not hyper-rec following the loss of RAD18 (Figure 1G). These data imply that the functional status of the MMR pathway is not predictive of the hyper-rec status of *RAD18*^−/−^ human cells.

To verify the differences in hyper-rec status between human cell lines, we assessed recombination utilizing two additional assays. First, we measured gene conversion, a type of nonreciprocal HR, in RAD18-proficient and RAD18-deficient cell lines using an established DR-GFP reporter assay (Figure 2A) [42]. In this assay, cells are transfected with a plasmid carrying a mutated *EGFP* gene containing an I-SceI endonuclease recognition site and a truncated *EGFP* fragment, along with an I-SceI expression plasmid. Transient expression of I-SceI generates a DSB in the mutated *EGFP* gene and requires gene conversion using the *EGFP* fragment to restore expression. Consistent with our SCE results, HR-mediated gene conversion was not reproducibly elevated in *RAD18*^−/−^ RPE-1 nor DLD1 cell lines (Figure 2B,C), but was increased approximately two-fold in HCT116 *RAD18*^−/−^ cells (Figure 2D). Importantly, this phenotype was rescued by the ectopic expression of FLAG-tagged RAD18 (Figure 2D). Next, we measured crossover recombination in RAD18-proficient and RAD18-deficient HCT116 and RPE-1 cells using a *phosphatidylinositol glycan class A* (*PIGA*) gene targeting assay (Figure 2G). *PIGA* encodes a protein that is essential for glycosylphosphatidyl inositol (GPI) anchor biosynthesis [51,52]. Membrane-associated GPI anchors can be detected by the Alexa 488-conjugated inactive aerolysin variant, FLAER [53]. To perform this assay, we first utilized CRISPR to functionally inactivate *PIGA* in RAD18-proficient and RAD18-deficient cells by inducing a frameshifting insertion-deletion (INDEL) into exon 6 (Figure 2E). *PIGA*-null cells were then transfected with *Cas9* mRNA, a CRISPR sgRNA targeting *PIGA* intron 5 to generate a DSB upstream of the exon 6 INDEL, and a donor plasmid containing the functional *PIGA* exon 6 sequence as a repair template. Successful targeted integration was then measured by quantifying the percentage of FLAER-positive cells. Consistent with our other assays, targeted integration was approximately two-fold higher in *RAD18*^−/−^ HCT116 cells (Figure 2F) but was not reproducibly increased in *RAD18*^−/−^ RPE-1 cells (Figure 2G). Taken together, these data confirm that RAD18-deficient HCT116 cells, but not RPE-1 nor DLD1 cells, are hyper-rec.

### 3.2. TLS Activity Is Moderately Reduced in RAD18^−/−^ and PCNA^K164/+^ Mutant HCT116 Cells

In light of our data showing that HCT116 cells become hyper-rec following the loss of RAD18, we wanted to evaluate whether RAD18 also retained its canonical role in TLS in this cell line. To examine the functional status of the TLS pathway, we measured the level of PCNA K164 ubiquitination and chromatin-bound DNA polymerase *eta* (Pol η) with or without UV irradiation (Figure 3A). A dose-dependent increase in PCNA ubiquitination was observed in wild-type cells, but not in *RAD18*^−/−^ cells. In addition, the recruitment of Pol η was moderately impaired in *RAD18*^−/−^ cell lines, suggesting that the activation of TLS was diminished following the loss of RAD18 in HCT116 cells. Next, we performed clonogenic survival assays with increasing doses of UV irradiation. Despite observing a trend towards increased UV sensitivity in *RAD18*^−/−^ cells, these data did not reach statistical significance (Figure 3B), consistent with previous data demonstrating that UV exposure did not reduce the viability of *RAD18*^−/−^ HCT116 cells [32].

Although our analyses of *RAD18*^−/−^ cells were consistent with RAD18′s published role in promoting TLS, the relatively weak phenotypes we observed prompted us to extend our experiments to PCNA K164 ubiquitination in HCT116 cells. To this end, we utilized CRISPR/Cas9-mediated gene targeting strategies either 5′ or 3′ of the codon for PCNA K164 at the endogenous locus to knock in the desired point mutation (c.491 A>G) accompanied by a diagnostic restriction enzyme recognition site (either EcoRI or XcmI; Figure 4A). In total, 195 candidate clones were screened by PCR amplification and restriction enzyme digestion with 10 clones showing complete digestion, implying homozygous integration of the diagnostic restriction enzyme recognition site. Examples for the 5′ strategy, (*PCNA^K164R/+^* clones 1A2 and 1D10) and the 3′ strategy (*PCNA^K164R/+^* clone 1B9) are shown in Figure 4B. Of the 10 candidate clones, Sanger sequencing determined that five were *PCNA^+/+^* and five were *PCNA^K164R/+^* (example trace files for clones 1A2 and 1B9 are shown in Appendix A). Thus, all HCT116 clones with biallelic targeting of *PCNA*, as defined by both alleles possessing the novel restriction site, were either heterozygous for the K164R coding mutation or wild-type. This was surprising, as we had employed a similar targeting strategy to successfully generate homozygous *PCNA^K164R/K164R^* RPE-1 cell lines [22]. These data suggest the possibility that, unlike in RPE1 cells, PCNA K164 may be essential in HCT116 cells.

To confirm our gene targeting results, we used CRISPR/Cas9 to introduce frameshifting INDELs into *PCNA* exon 2 in heterozygous *PCNA^K164R^*^/*+*^ mutant cells. Because *PCNA* is essential, successful targeting should result in viable clones that are either *PCNA^+^*^/*−*^ or *PCNA^K164R^*^/*−*^ whereas *PCNA^−^*^/*−*^ clones would be inviable. Moreover, we predicted that all viable clones would be *PCNA^+^*^/*−*^ if the K164R coding mutation was lethal. To validate this strategy, PCR amplicons were generated using primers upstream of the Cas9 cleavage site in exon 2 and downstream of the K164 codon in exon 5 (Appendix A) from the targeted population and were cloned into a pCR2.1-TOPO vector to isolate single alleles that were analyzed by Sanger sequencing. These data demonstrated that frameshifting INDELs could successfully knock out either the wild-type or *PCNA^K164R^* allele (Appendix A). Intriguingly, however, once the polyclonal population was subcloned not only did we not obtain viable *PCNA^K164R^*^/*−*^ clones, but we were also unable to isolate heterozygous *PCNA^+^*^/*−*^ clones. These data are consistent with the notion, but do not unequivocally prove, that PCNA K164 is essential and that *PCNA* may even be haplolethal in HCT116 cells.

To confirm *PCNA* haplolethality, we attempted to generate heterozygous *PCNA* mutants by targeting exon 2 in wild-type cells. We monitored locus-specific INDELs within targeted cell populations using PCR analysis spanning the Cas9 cleavage site followed by deconvolution of Sanger sequencing traces using the Tracking of Indels by Decomposition (TIDE) algorithm [54]. As a control, we targeted *minichromosome maintenance protein 10 (MCM10)* in parallel. Like *PCNA*, *MCM10* is an essential gene; however, heterozygous HCT116 *MCM10* mutants are viable [55]. The initial INDEL frequency at the *PCNA* and *MCM10* loci were comparable at ~67% and ~50%, respectively (Figure 4C). Yet, over the course of seven days, the INDEL frequency at the *PCNA* locus decreased dramatically (to just 4%) in comparison to the *MCM10* locus which remained stable (Figure 4C).

Altogether, our inability to generate heterozygous *PCNA^+^*^/*−*^ or *PCNA^K164R^*^/*−*^ cell lines implies that *PCNA* is haplolethal in HCT116 cells. Therefore, subsequent experiments focused on the characterization of the heterozygous *PCNA^K164R^*^/*+*^ cell lines 1A2 and 1D10.

First, we analyzed PCNA ubiquitination following UV irradiation. This treatment caused a dose-dependent increase in PCNA ubiquitination in both wild-type and *PCNA^K164R^*^/*+*^ cells, with 1.5- to 2-fold higher levels in wild-type cells than in *PCNA^K164R^*^/*+*^ mutants (Figure 4D). It is notable that the heterozygous *PCNA^K164R^*^/*+*^ mutants displayed statistically significant sensitivity over a lower range of UV exposure (5, 7, and 10 J; Figure 4E) in comparison to insensitive *RAD18^−^*^/*−*^ mutants over a higher dose range (5, 10, and 20 J; Figure 3B). Furthermore, phosphorylated replication protein A (pRPA32) and phosphorylated histone H2AX (γH2AX) were significantly elevated in irradiated *PCNA^K164R^*^/*+*^ cells in comparison to wild-type cells (Figure 4D). Consistent with the modest reduction in PCNA K164 ubiquitination, *PCNA^K164R^*^/*+*^ cells showed slight, but statistically significant, sensitivity to UV irradiation (Figure 4E) and methyl methanesulfonate (MMS; Appendix A). Finally, we performed a *hypoxanthine phosphoribosyltransferase (HPRT)* mutagenesis assay in *PCNA^K164R^*^/*+*^ and *RAD18^−^*^/*−*^ mutants following UV irradiation to indirectly measure error-prone synthesis (i.e., TLS) (Figure 4F). HPRT catalyzes the processing of the purine analog 6-thioguanine (6-TG) to 6-thioguanine 5′-monophosphate, which is incorporated into DNA and results in cytotoxicity [56]. UV-induced TLS at the *HPRT* locus can introduce point mutations that disrupt the enzymatic activity of HPRT and promote cell survival in the presence of 6-TG. Therefore, we measured the frequency of 6-TG-resistant colonies as an indirect readout for TLS activity. UV irradiation led to a clear, although statistically insignificant, increase in 6-TG-resistant HCT116 wild-type colonies but did not increase the number of 6-TG-resistant colonies in *RAD18^−^*^/*−*^ or in *PCNA^K164R^*^/*+*^ cell lines (Figure 4G). Taken together, our analyses of *RAD18^−^*^/*−*^ and *PCNA^K164R^*^/*+*^ cell lines are consistent with the requirement for RAD18-catalyzed PCNA K164 mono-ubiquitination to achieve robust levels of TLS in HCT116 cells. Notably, this did not correlate with significantly decreased cell survival following UV exposure, consistent with previous data in HCT116 cells [32], but contrary to observations in RAD18-deficient chicken DT40 or mouse embryonic stem cells [29,31,33].

### 3.3. Hyper-Recombination Is Linked to PCNA K164 Ubiquitination and Not SUMOylation

The results of our analyses of HCT116 *RAD18^−^*^/*−*^ and *PCNA^K164R^*^/*+*^ mutant cell lines are reminiscent of previous studies of RAD18 and PCNA K164 function in avian DT40 cells, which found that loss of either caused hyper-recombination and reduced TLS. Based on their data, Simpson et al. [29] proposed that RAD18 and PCNA each play a role in suppressing SCE and that hyper-recombination may be a compensatory mechanism to promote cell survival when DNA gaps accumulate due to compromised TLS. Thus, we wondered if the hyper-rec phenotype in HCT116 cells was caused by the inability to post-translationally modify all PCNA proteins at K164, and whether this was specifically linked to ubiquitination but not SUMOylation. To address this, we generated clones that had reverted the K164R coding mutation to wild-type at the endogenous locus, similar to what we have described for RPE-1 cells [22]. Western blot analyses of these cell lines left untreated or exposed to UV irradiation confirmed that the level of PCNA ubiquitination was reduced in *PCNA^K164R^*^/*+*^ cells and was restored in reverted cell lines (Figure 5A). Next, we evaluated recombination at the gross chromosomal level using the SCE assay and specifically at telomeres by measuring telomeric SCEs (t-SCEs). We were motivated to examine telomeres because they contain repetitive, hard-to-replicate DNA sequences and are classified as fragile sites [57]. Based on these characteristics, we hypothesized that recombination might be particularly elevated at telomeres. To facilitate this analysis, we generated *PCNA^K164R^*^/*+*^ and *PCNA^+^*^/*+*^ reverted cell lines expressing “super-telomerase” [40], a strategy to lengthen telomeres that we have successfully employed to facilitate the evaluation of telomere replication and maintenance in HCT116 cell lines [55]. Clones with similarly lengthened telomeres, as measured by a telomere restriction fragment length assay, were selected for further analyses (Appendix A).Quantification of genomic SCEs indicated that these events were increased approximately two-fold in *PCNA^K164R^*^/*+*^ mutants, and that SCEs were restored to wild-type levels in reverted cells (Figure 5B). Interestingly, our quantification of t-SCEs showed that recombination was more significantly elevated at telomeres in *PCNA^K164R^*^/*+*^ cells in comparison to genome-wide SCEs, and that telomere-specific recombination was also rescued in reverted cells (Figure 5C,D). Finally, given that PCNA K164 SUMOylation suppresses HR [16,18], we asked whether the observed increase in recombination in *PCNA^K164R^*^/*+*^ cells was due to decreased SUMOylation. Because SUMOylated PCNA is particularly difficult to detect in human cells [17], we addressed this question using siRNA knockdown of UBC9, the E2 conjugating enzyme needed for this modification [16]. This approach successfully depleted UBC9 (Figure 4E) but did not increase t-SCEs in wild-type, *PCNA^K164R^*^/*+*^, or reverted cells (Figure 4F). Although we cannot rule out the possibility that a low level of residual K164 SUMOylation remains and is sufficient to affect the rate of t-SCEs, our data are consistent with the notion that the increased recombination is related to reduced RAD18-mediated PCNA K164 ubiquitination.

## 4. Discussion

### 4.1. Human RAD18 Suppresses Hyper-Recombination in Specific Cell Lines

In general, mammalian cells strictly regulate HR to avoid genetic alterations which may contribute to cellular transformation and carcinogenesis [58,59]. Here, we investigated the effect of RAD18 on HR-mediated repair mechanisms in three cell lines. We report that in the absence of RAD18, HCT116 cells upregulate HR-dependent processes including SCE (Figure 1H,J), plasmid-templated gene conversion (Figure 2D), and CRISPR/Cas9-mediated targeted integration (Figure 2F). Conversely, loss of RAD18 did not lead to hyper-recombination in RPE-1 nor DLD1 cells (Figure 1F,G and Figure 2B,C,G). Because HR-mediated repair and MMR are linked [48] and HCT116 cells are MMR-deficient, we wondered whether the MMR status affected hyper-recombination in RAD18-deficient cells. Contrary to this notion, MMR-deficient DLD1 and RPE-1 cells, which are naturally deficient or were engineered to be so using CRISPR/Cas9 gene targeting, respectively, were not hyper-rec (Figure 1G, Figure 2C and Appendix A). Although these data do not exclude the possibility that an MMR deficiency affects the hyper-rec phenotype in HCT116 *RAD18^−^*^/*−*^ mutants, our results imply that the hyper-rec status of human cells is not solely dictated by whether or not the MMR pathway is functional. We propose that there are specific genetic or cellular contexts in which RAD18 is necessary for the suppression of hyper-recombination. One specific example from the published literature is the presence of mutant *meiotic recombination 11* (*MRE11*) allele in HCT116 cells [60]. Transformation status and, presumably, differences in checkpoint function are also notable differences between RPE-1, DLD1, and HCT116 cell lines. Indeed, the dynamics of a p53 response to DNA damage can vary greatly across cell lines [61] and it was recently proposed that p53 levels direct the hierarchy of DDT pathway usage in some cell types [62]. Future experiments exploring these differences may identify additional factors that dictate the hyper-rec status of RAD18^−/−^ human cells. Our data, discussed below, suggest that these phenotypes are linked to PCNA K164 ubiquitination.

### 4.2. PCNA Is Haplolethal, and PCNA K164 May Be Essential, in HCT116 Cancer Cells

To understand whether the hyper-rec phenotypes in *RAD18* null HCT116 cells were linked to PCNA K164 ubiquitination, we attempted to generate homozygous *PCNA^K164R^*^/*K164R*^ cell lines. Unexpectedly, we were unsuccessful in generating not only *PCNA^K164R^*^/*K164R*^ mutants but *PCNA^K164R^*^/*−*^ and *PCNA^+^*^/*−*^ cell lines as well. Our data demonstrate that both *PCNA* alleles can be targeted simultaneously (Figure 4A,B and Appendix A), implying that the inability to generate homozygous clones was not simply due to poor targeting efficiency. In agreement with these observations, following CRISPR/Cas9 targeting of *PCNA* exon 2 in a wild-type HCT116 population, cells carrying INDELs were rapidly lost (Figure 4C). However, targeting *MCM10* in parallel showed a stable INDEL frequency in the population over the same time span, consistent with the fact that heterozygous *MCM10* HCT116 cells are viable [55]. Overall, these data suggest that *PCNA* is haplolethal in HCT116 cells. This was surprising given our previous success in generating *PCNA^K164R^*^/*K164R*^ and *PCNA^K164R^*^/*−*^ mutant RPE-1 cell lines [22,63]. Notably, RPE-1 cells do not become hyper-rec following the loss of RAD18 (Figure 1F and Figure 2B,G) and are viable as homo- or hemizygous *PCNA^K164R^* mutants [22,63]. Conversely, HCT116 cells are hyper-rec following the loss of RAD18 and our data suggest that they cannot tolerate homo- or hemizygous *PCNA^K164R^* expression. These observations indicate that the genetic background of each cell type affects these phenotypes and that future experiments could provide valuable insights into the link(s) between RAD18 and PCNA K164 in different gap-filling DNA synthesis and repair pathways.

### 4.3. Suppression of Hyper-Recombination by RAD18 Is Linked to PCNA K164 Ubiquitination

The requirement for RAD18-mediated PCNA K164 ubiquitination has evolved from yeast, where this activity is essential for TLS [6,64], to be more complex in mammalian cells. For instance, there is approximately 25% residual TLS activity in *PCNA^K164R^*^/*K164R*^ and *RAD18^−^*^/*−*^ mouse embryonic fibroblasts [65], and *PCNA^K164R^*^/*K164R*^ HEK293T cells [63], suggesting that some TLS polymerase recruitment occurs independently of PCNA K164 ubiquitination. Our analyses of TLS in hyper-rec *RAD18^−^*^/*−*^ HCT116 mutants found that although PCNA K164 ubiquitination and chromatin association of the TLS polymerase η were reduced (Figure 3A), these cells did not show a strong corresponding reduction in viability following UV irradiation (Figure 3B). Exposure of *PCNA^K164R^*^/*+*^ cells to either UV or MMS showed a modest but statistically significant decrease in clonogenic survival (Figure 4E and Appendix A). Finally, using an *HPRT* mutagenesis assay, we found that *RAD18*^−/−^ and *PCNA^K164R^*^/*+*^ mutants did not display robust TLS (Figure 4G). These data are consistent with reduced gap-filling synthesis in these cell lines and, in addition, it is plausible that other post-replicative synthesis pathways are also affected by diminished RAD18-catalyzed mono-ubiquitination of PCNA K164.

We propose that hyper-recombination in *RAD18*^−/−^ HCT116 cells compensates for reduced gap-filling DNA synthesis and speculate that this compensation maintains viability. Consistent with this idea, *PCNA^K164R^*^/*+*^ cells were also hyper-rec, as indicated by significantly increased genome-wide and telomere-specific SCEs (Figure 5B,D). Importantly, both phenotypes were rescued following the reversion of the K164R coding mutation at the endogenous locus back to wild-type. One recurring question in investigations of PCNA K164 function(s) is whether phenotypes resulting from the K > R mutation are related to ubiquitination, SUMOylation, or both. Relevantly, knockdown of UBC9 to reduce SUMOylation had no impact on t-SCEs, suggesting that SUMOylation of K164 does not suppress HR, unlike observations of PCNA SUMOylation in yeast [16,19]. These data highlight the importance of K164 ubiquitination and suggest that RAD18 is the key E3 ligase, although we cannot rule out contributions of additional ligases such as CRL4(Cdt2) or BRCA1/BARD1 [66,67,68]. Our data are reminiscent of those reported by Simpson et al. in *rad18^−^*^/*−*^ or *pcna^K164R^* mutant DT40 mutant cell lines [29]. Similarly, we speculate that in HCT116 cells HR is increased to compensate for reduced gap-filling repair synthesis (Figure 6) and that this compensatory mechanism is not required in certain cell types (e.g., RPE-1 and DLD1 cells). Increased activity of an alternative E3 ligase to promote PCNA K164 ubiquitination and TLS or the upregulation of an alternative gap-filling synthesis pathway might explain why *RAD18*^−/−^ RPE-1 and DLD1 cells do not become hyper-rec. Generally, we propose that when cells experience replication stress, gap-filling repair synthesis can be completed by a variety of pathways, but that cell-specific hierarchies dictate the preferred pathway(s). Future studies are needed to address the details of these hierarchies within different organisms or cell types.

## 5. Conclusions

Our data demonstrate that inactivation of RAD18 results in hyper-recombination in HCT116 but not RPE-1 nor DLD1 human cell lines. The HCT116 hyper-rec phenotype is linked to RAD18-mediated PCNA K164 ubiquitination, but not the sumoylation of the same residue. We speculate that HR is upregulated to prevent replication stress and DNA damage that may result from the inability to engage RAD18-dependent gap-filling DNA synthesis pathways. In conjunction with the present literature, our data support the notion that eukaryotic cells preferentially engage specific pathways to reduce replication stress and maintain genome stability and that these preferences are intrinsic and must be experimentally defined for each cell type.

## Figures and Tables

**Figure 1 biomolecules-15-00150-f001:**
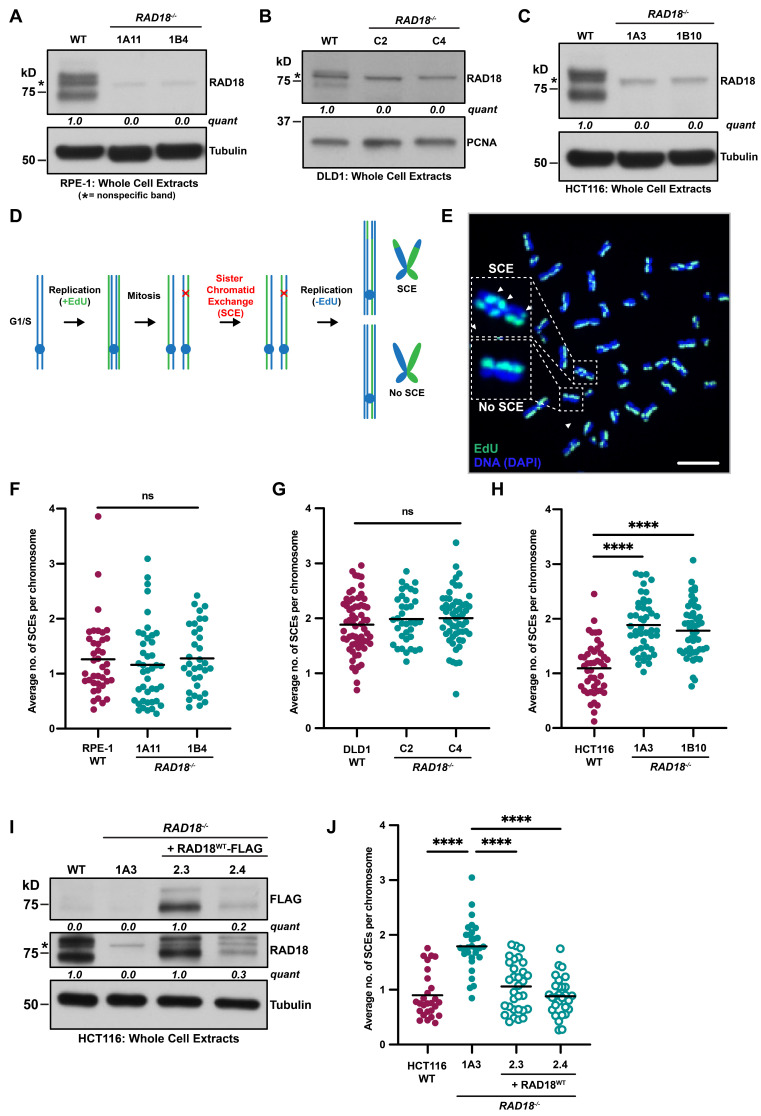
RAD18-mediated suppression of recombination is cell line specific. (**A**–**C**) Western blot analyses confirm the loss of RAD18 in two independent RPE-1 (**A**), DLD1 (**B**), and HCT116 (**C**) *RAD18*^−/−^ clones. The top band detected by the anti-RAD18 antibody is mono-ubiquitinated RAD18 and the bottom band is unmodified RAD18 [45]. The asterisk indicates a nonspecific band recognized by the anti-RAD18 antibody. Quantification is based on densitometry using FIJI normalized to the loading control and WT lane, minus signal from the nonspecific RAD18 band. (**D**) Schematic of the SCE assay. (**E**) Example metaphase spread from an HCT116 *RAD18*^−/−^ cell used for SCE analyses. One SCE is scored each time there is a switch from green-labeled DNA to blue-labeled DNA (or vice versa) on a sister chromatid. The scale bar is 10 µm. (**F**–**H**) The average number of SCEs per chromosome in RPE-1 (**F**), DLD1 (**G**), or HCT116 (**H**) cell lines. Each dot represents one metaphase spread across two biological replicates. Black lines indicate mean values and significance was calculated using a Mann–Whitney test with **** > 0.0001 and ns = not significant. (**I**) Western blot analysis confirming complementation of HCT116 *RAD18*^−/−^ clone 1A3 with wild-type FLAG-tagged RAD18 in two independent clones (2.3 and 2.4). The asterisk indicates a nonspecific band recognized by the anti-RAD18 antibody. Quantification is based on densitometry using FIJI normalized to the loading control and WT lane (RAD18) minus signal from the nonspecific RAD18 band or normalized to the loading control and 2.3 lane (FLAG). (**J**) Average number of SCEs per chromosome in HCT116 cell lines. Each dot represents one metaphase spread within a single biological replicate. Black lines indicate mean values and significance was calculated using a Mann–Whitney test with **** > 0.0001. Original western blot images can be found in Appendix A.

**Figure 2 biomolecules-15-00150-f002:**
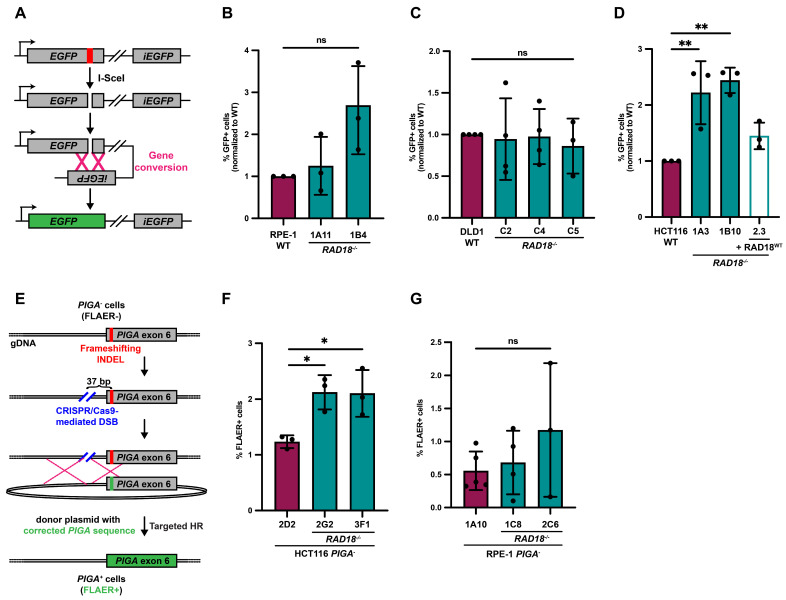
HCT116 *RAD18^−^*^/*−*^ cells exhibit hyper-recombination phenotypes. (**A**) Schematic of the DR-GFP gene conversion reporter assay. (**B**–**D**) Results of the DR-GFP gene conversion assay. The average percentage of GFP^+^ cells normalized to wild-type for RPE-1 (**B**), DLD1 (**C**) or HCT116 (**D**) cell lines. For each cell line n = at least 9 across at least three biological replicates. Each column is normalized to the baseline, defined as the mean of all WT values. The error is indicated as standard deviation and significance was calculated using Dunnett’s one-way ANOVA tests, with ** > 0.01 and ns = not significant. (**E**) Schematic of *PIGA* gene targeting assay. (**F**,**G**) Results of the *PIGA* gene targeting assay. The average percentage of FLAER-positive (FLAER^+^) cells in HCT116 (**F**) or RPE-1 (**G**) cell lines. For each cell line n = 9 wells across three biological replicates. The error is indicated as standard deviation and significance was calculated using Dunnett’s one-way ANOVA tests, with * > 0.05 and ns = not significant.

**Figure 3 biomolecules-15-00150-f003:**
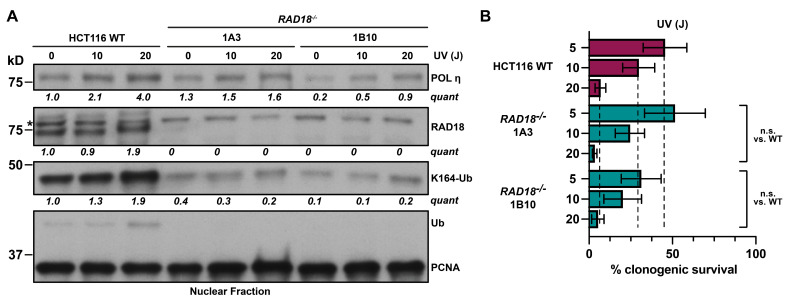
TLS activation is reduced in *RAD18^−^*^/*−*^ cells without significantly decreased viability following UV irradiation. (**A**) Western blot analysis of POL η, RAD18, PCNA, and PCNA ubiquitinated at K164 levels in nuclear fractions from HCT116 wild-type and *RAD18^−^*^/*−*^ cell lines with or without preceding UV irradiation (10 or 20 J). The asterisk indicates a nonspecific band recognized by the anti-RAD18 antibody. Quantification is based on densitometry using FIJI normalized to the loading control and WT lane. (**B**) Clonogenic survival of HCT116 wild-type and two *RAD18*^−^^/−^ cell lines following UV irradiation at the indicated dosages (5, 10, or 20 J). Error is indicated as SEM and significance was calculated using an unpaired, two-tailed Student’s *t*-test across four biological replicates. n.s. = not significant. Original western blot images can be found in Appendix A.

**Figure 4 biomolecules-15-00150-f004:**
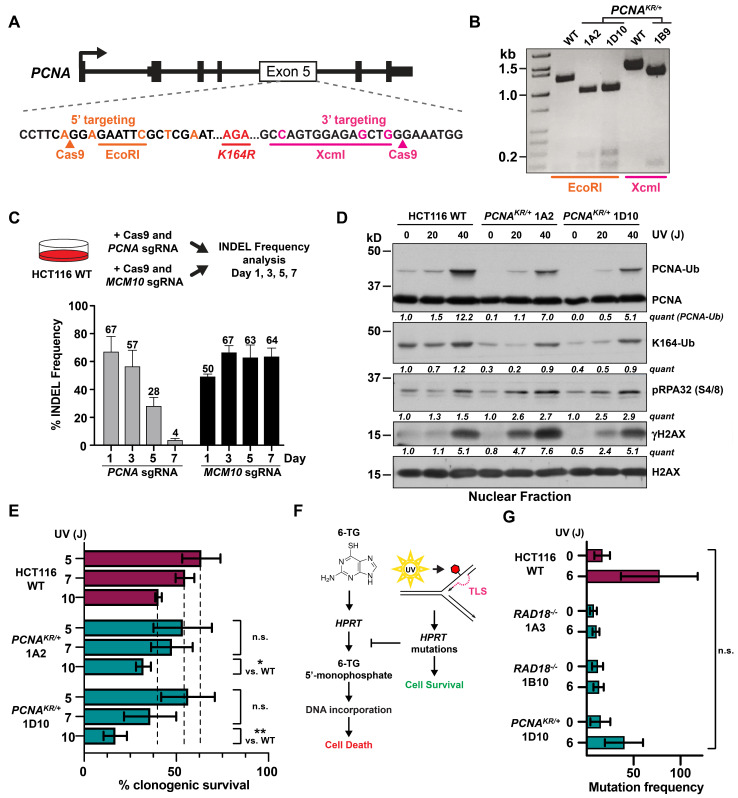
Generation and characterization of HCT116 *PCNA^K164R^*^/*+*^ mutant cell lines. (**A**) Schematic of the *PCNA* exon 5 targeting strategies used to generate HCT116 *PCNA^K164R^*^/*+*^ mutant cell lines. (**B**) Representative results from the diagnostic PCR followed by EcoRI or XcmI restriction enzyme digestion to identify successful targeting of *PCNA* exon 5. Expected products for the 5′ targeting strategy include the following: not targeted (WT) 180 bp, 1246 bp; biallelic targeting (*PCNA^K164R^*^/*+*^ 1A2, 1D10) 180 bp, 992 bp, and 254 bp. Expected products for the 3′ targeting strategy include the following: not targeted (WT) 1460 bp; biallelic targeting (*PCNA^K164R^*^/*+*^ 1B9) 1260 bp and 166 bp. (**C**) (Top) Schematic of an experiment to determine HCT116 *PCNA* haplolethality. (Bottom) The percent INDEL frequency as measured by TIDE analysis in HCT116 populations following *PCNA* exon 2 or *MCM10* exon 3 CRISPR/Cas9 targeting, respectively, over a seven-day time course. Error is indicated as standard deviation representing two biological replicates. (**D**) Western blot analyses of PCNA, ubiquityl-PCNA (K164), phospho-RPA32 (S4/8), and γH2AX, with or without UV treatment (20 or 40 J), with histone H2AX as the loading control, in nuclear fractions isolated from HCT116 cell lines. Quantification is based on densitometry using FIJI normalized to the loading control and WT lane. (**E**) Clonogenic survival of HCT116 wild-type and two *PCNA^K164R^*^/*+*^ cell lines following UV irradiation at the indicated dosages (5, 7, or 10 J). Error is indicated as standard deviation and significance was calculated using a two-tailed students *t*-test with * > 0.05; ** > 0.01 across at least three biological replicates. (**F**) Schematic of the *HPRT* survival assay. (**G**) Average mutation frequency at the *HPRT* locus in untreated or UV-treated (6 J) HCT116 cell lines. Error is indicated as SEM and significance was calculated using Sidak’s multiple comparison test across three biological replicates. n.s. = not significant. No comparisons were statistically significant. Original western blot images can be found in Appendix A.

**Figure 5 biomolecules-15-00150-f005:**
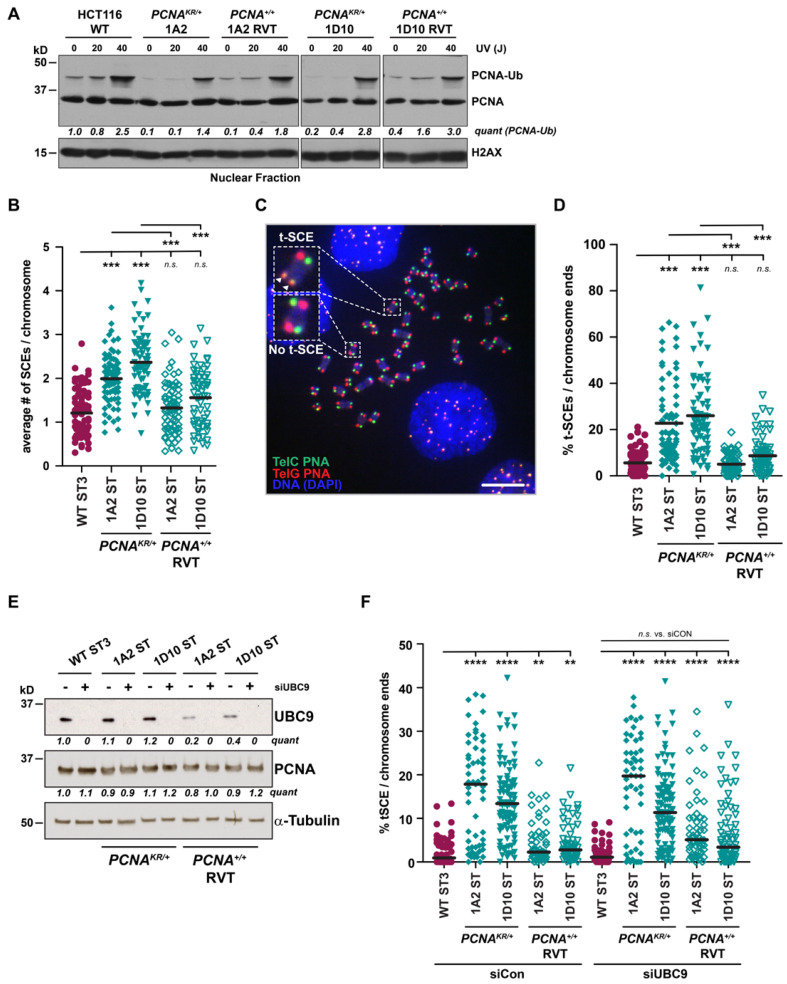
Reduced PCNA K164 ubiquitination leads to increased global and telomeric recombination. (**A**) Western blot analysis of PCNA levels in nuclear fractions with or without UV treatment (20 or 40 J), with histone H2AX as the loading control, in HCT116 wild-type, *PCNA^K164R^*^/*+*^, and *PCNA^+^*^/*+*^ reverted cell lines. Quantification is based on densitometry using FIJI normalized to the loading control and WT lane. (**B**) Quantification of the average number of SCEs per chromosome in HCT116 wild-type, *PCNA^K164R^*^/*+*^, and *PCNA^+^*^/*+*^ reverted ST cell lines. Black lines indicate mean values and statistical significance was calculated using Kruskal–Wallis with Dunn’s multiple comparison test with *** < 0.001; across two biological replicates. (**C**) Representative metaphase spread for spontaneous telomeric SCEs (t-SCEs). Examples of a t-SCE event and chromosomes without t-SCE are highlighted. The scale bar is 20 µm. (**D**) Quantification of the percent t-SCEs per chromosome ends in HCT116 wild-type, *PCNA^K164R^*^/*+*^, and *PCNA^+^*^/*+*^ reverted ST cell lines. Black lines indicate mean values and statistical significance was calculated using Kruskal–Wallis with Dunn’s multiple comparison test *** < 0.001; across two biological replicates. (**E**) Western blot analysis of UBC9 and PCNA levels in HCT116 wild-type, *PCNA^K164R^*^/*+*^, and *PCNA^+^*^/*+*^ reverted ST cell lines following a 72 h treatment with siControl or siUBC9, with tubulin as a loading control. Quantification is based on densitometry using FIJI normalized to the loading control and WT lane. (**F**) Quantification of the percent t-SCEs per chromosome ends in HCT116 wild-type, *PCNA^K164R^*^/*+*^, and *PCNA^+^*^/*+*^ reverted ST cell lines following a 72 h treatment with siControl or siUBC9. Black lines indicate mean values and statistical significance was calculated using Kruskal–Wallis with Dunn’s multiple comparison test ** < 0.01, **** < 0.0001, n.s. = not significant; across two biological replicates. Original western blot images can be found in Appendix A.

**Figure 6 biomolecules-15-00150-f006:**
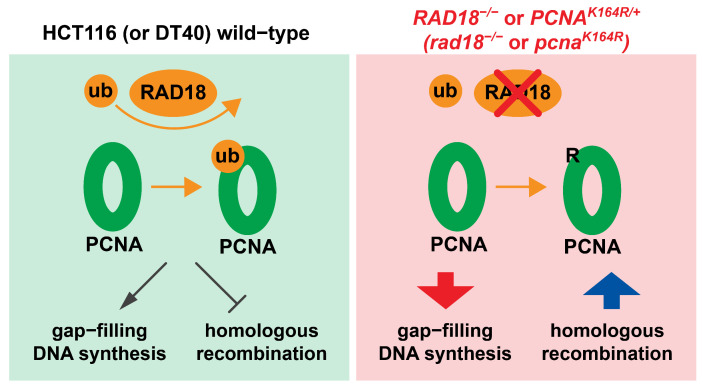
Suppression of hyper-recombination by RAD18 is linked to PCNA K164 ubiquitination. In wild-type HCT116 and DT40 cells, the balance between gap-filling DNA synthesis and HR is regulated by the activity of RAD18 and the level of PCNA K164 ubiquitination. In these cell lines, loss of RAD18 results in a significant increase in HR. We speculate that this occurs to compensate for the deficiency in gap-filling DNA synthesis and improve viability.

## Data Availability

The data presented in this study are included in the article and Appendix A. The raw data supporting the conclusions of this article will be made available by the authors upon reasonable request.

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
