# Peer review of "Cell Type Specific Suppression of Hyper-Recombination by Human RAD18 Is Linked to Proliferating Cell Nuclear Antigen K164 Ubiquitination"

_biomolecules, 2025, doi:10.3390/biom15010150_

Round 1

Reviewer 1 Report

Comments and Suggestions for Authors

In this manuscript, Rogers et al. report that Rad18 knockout causes hyper-recombinogenic phenotypes in HCT116 cells, but not in hTERT RPE-1 and DLD1. While RAD18 is a multi-function protein, they attribute this hyper-recombination in HCT116 to a defect in PCNA K164 ubiquitination, but not SUMOylation. Additionally, the authors report that PCNA is haploinsufficient and PCNA K164 is essential in HCT116.

This study reports an important observation that consequences of RAD18 deficiency is cell-type specific. Experiments were performed rigorously (the hyper-rec phenotypes are supported by using three different assays including sister chromatid exchange, DR-GFP, and gene targeting), and the presented data are high quality. However, some of the conclusions appear to be overstated, and the model presented at the end is misleading.

My specific comments are listed below for the authors to consider during the revision.

Major points:

1.  Figure 3: The authors show that TLS activation is reduced in HCT116 RAD18-/- cells. It would be informative to examine whether RPE-1 RAD18-/- and DLD-1 RAD18-/- cells are TLS deficient, as the lack of hyper-rec phenotypes in these cells could be explained if gap-filling DNA synthesis is less dependent on Rad18 in RPE-1 and DLD1 than in HCT116.

2.  Lines 658-659: Although the authors “propose that hyper-recombination in RAD18-/- HCT116 cells compensates for reduced gap-filling DNA synthesis to maintain viability”, whether the hyper-rec events promote the viability of HCT116 RAD18-/- cells has not been demonstrated. This applies to Lines 684-686 as well. It is recommended to either revise these statements to accurately reflect the data or rephrase them so that it is clear that they are speculations.

3.  Figure 6, lines 670-674, and lines 690-692: The model might be misleading. The presented data only support differential consequences of RAD18 knockout between cell lines. This cannot be interpreted as cell-type specific response to replication stress as implied in this model.

Minor points:

1. Line 153: Is K164/+ supposed to be K164R/+?

2. Line 376: 10 μM should be 10 μm

Reviewer 2 Report

Comments and Suggestions for Authors

The manuscript describes differences on hyperrrecombination of the lack of RAD18 and PCNA-K164R on different cell lines. The data here presenting is intriguing and relevant to the field. However, other ubiquitin E3 enzymes have been described to ubiquitinate PCNA on K164, including CRL4(cdt2) (Terai et al. 2010 Mol Cell  https://doi.org/10.1016/j.molcel.2009.12.018) and recently BRCA1/BARD1 (Salas-Lloret et al 2024 Nature Communs). Specially the latest, as it is involved in homologous recombination should be mentioned and commented alongside the results.

Reviewer 3 Report

Comments and Suggestions for Authors

This manuscript describes the distinct usage of DNA damage tolerance (DDT) pathways in different human cell lines (HCT116, DLD1, and hTERT RPE1-1). Unlike yeast, many studies on human cell lines are conducted on backgrounds with different genetic variations, making it challenging to compare results across cell lines. In this context, the results of this study are significant, as they highlight that the requirements for DDT pathways vary among cell types. Particularly, the study demonstrated that in HCT116 cells, these pathways play a more influential role in suppressing homologous recombination compared to other cell lines.

In my opinion, the overall presentation of the data is sound, and most of the results are valuable to researchers in the relevant field. However, I believe some parts of the manuscript require more careful description (as outlined below):

-  Genome editing of PCNA-K164R

The authors did not sufficiently provide evidence that biallelic PCNA-K164R mutations result in cell inviability. In Figure 4C, they demonstrated that one allele of PCNA (wt or K164R) is not sufficient for cell growth and this haplolethality is independent of K164R mutation. On the other hand, a systematic experiment to verify the consequences of biallelic PCNA-K164R mutations has not been conducted.  Therefore, the authors should avoid making definitive conclusions on this point. Specifically, the descriptions in lines 443–444, 469–470, and 637, among others, should be revised or omitted.

- Terminology: "on-the-fly gap-filling synthesis"

The term "on-the-fly gap-filling synthesis" is frequently used in the manuscript, but its precise meaning is unclear. Single-stranded DNA gaps are likely generated due to repriming synthesis downstream of replication forks, meaning that gap filling should occur separately from ongoing replication fork activity. Thus, this terminology does not seem appropriate for describing events directly associated with replication forks.

- Figure 6: Representation of post-replicative gap filling

In Figure 6, the authors illustrate "post-replicative gap filling" and "on-the-fly gap-filling synthesis" in regions where two replication forks are merging or terminating. However, to my knowledge, there is no evidence that RAD18 or PCNA ubiquitylation contributes to the termination of replication forks. Additionally, it is unclear whether HR-dependent DSB repair occurs in these regions. Therefore, I suggest that the authors either remove this figure or modify it appropriately to reflect existing evidence.
